# Asymmetric migration decreases stability but increases resilience in a heterogeneous metapopulation

Anurag Limdi [1,4], Alfonso Pérez-Escudero [1,5], Aming Li [1,2,3,6] & Jeff Gore [1]

Many natural populations are spatially distributed, forming a network of subpopulations linked by migration. Migration patterns are often asymmetric and heterogeneous, with important consequences on the ecology and evolution of the species. Here we investigate experimentally how asymmetric migration and heterogeneous structure affect a simple metapopulation of budding yeast, formed by one strain that produces a public good and a non-producer strain that benefits from it. We study metapopulations with star topology and asymmetric migration, finding that all their subpopulations have a higher fraction of producers than isolated populations. Furthermore, the metapopulations have lower tolerance to challenging environments but higher resilience to transient perturbations. This apparent paradox occurs because tolerance to a constant challenge depends on the weakest subpopulations of the network, while resilience to a transient perturbation depends on the strongest ones.

[1] Physics of Living Systems, Department of Physics, Massachusetts Institute of Technology, Cambridge, MA 02139, USA. [2] Center for Systems and Control, College of Engineering, Peking University, Beijing 100871, China. [3] Center for Complex Network Research and Department of Physics, Northeastern University, Boston, MA 02115, USA. [4] Present address: Department of Molecular and Cellular Biology, Harvard University, Cambridge, MA 02138, USA. [5] Present address: Centre de Recherches sur la Cognition Animale, Centre de Biologie Intégrative, Université de Toulouse, CNRS, UPS, Toulouse Cedex, France. [6] Present address: Chair of Systems Design, ETH Zürich, Weinbergstrasse 56/58, Zürich CH-8092, Switzerland. These authors contributed equally: Anurag Limdi, Alfonso Pérez-Escudero. Correspondence and requests for materials should be addressed to J.G. (email: gore@mit.edu)

Natural populations are spatially distributed, often in a way in which neighboring populations are linked to each other by migration. These complex populations are called metapopulations[1–6] or, when they contain several coexisting species, metacommunities[7,8]. Natural metapopulations and metacommunities often have heterogeneous topology (some subpopulations have more neighbors than others) and asymmetric migration patterns (the net flow of individuals between any two subpopulations can be non-zero). Extreme examples are dendritic networks, in which several nodes of each level connect to a single node in the next one[9]. For example, river basins give rise to dendritic metacommunities, with all tributaries connected to their parent river (and with highly asymmetric migration due to water currents)[9–15]. Besides dendritic networks, many natural metacommunities have heterogeneous connectivity and asymmetric migration patterns[16,17].

A major challenge is understanding how the structure of metapopulations and metacommunities influences the ecology and evolution of the involved species. Spatial structure usually increases the biodiversity of metacommunities[8,18–23], and may enhance particular interactions such as cooperation[24–27]. It may also have profound effects on the ability of a species to survive environmental deterioration and transient perturbations. This question has been extensively studied theoretically[28–33] and experimentally in designed[19,34–36] and real-world[20,37] metapopulations and metacommunities. Given the complexity of these systems, results are mixed: Depending on the conditions, spatial structure may increase[20,28–30,37] or decrease[29–32] the system's ability to survive in a challenging environment. Results from stability theory in dynamical systems may help identify the main factors determining each outcome[38]. For example, well-mixed populations and communities have been shown to cross a tipping point as the environment deteriorates, leading to a sudden collapse of the population rather than a smooth decline towards extinction[39–41]. This detailed understanding allows predicting how different factors affect the stability of the populations, and has also helped to demonstrate that generic indicators such as critical slowing down can predict the collapse of the system[38,39,41–43]. Extending this approach to metapopulations requires taking into account the effect of migration on the density of the subpopulations. This effect is often neglected, as the most studied effect of migration is to propagate species to locations where they are not present[5,6,44–46]. Yet, in many cases, migration may be strong enough to have a significant effect in the density of the subpopulations—the so-called mass effects[7,47], which in turn may determine their survival and composition (via density-dependent selection).

Here we address these questions experimentally, taking advantage of the high-throughput and short generation times of microbial microcosms, which allow us to study metapopulations over hundreds of generations. By studying metapopulations with star topology, we find that asymmetric migration increases the fraction of producers in all nodes of the metapopulation. It also makes the metapopulation less stable (i.e. less capable of surviving in challenging environments), but more resilient to transient perturbations. This apparent paradox happens because stability depends on the weakest subpopulations, while resilience depends on the strongest ones.

## Results

**Experimental system**. We studied two strains of budding yeast (*Saccharomyces cerevisiae*) growing on sucrose. These cells cannot metabolize sucrose directly; one of the strains (the producer) produces an enzyme that breaks down sucrose into glucose and fructose, which can be metabolized by the cells. This reaction

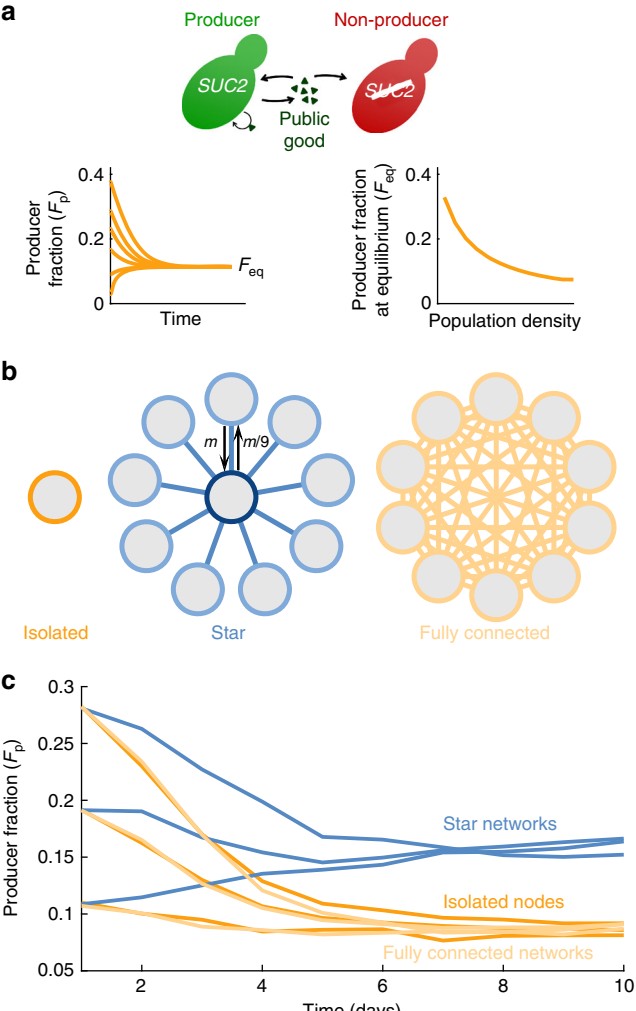

**Fig. 1** Heterogeneous metapopulation structure increases the fraction of producers. **a** Schematic of the public-goods game. Producers (green) produce a public good (triangles), keeping a small fraction (~1%) for themselves and releasing the rest, which also benefits non-producers (red). Left inset: Illustration of the time evolution of the fraction of producers ($F_p$) in a well-mixed population. Right inset: Illustration of the dependency of the equilibrium fraction of producers ($F_{eq}$) on the density of a well-mixed population. Both illustrations were computed using the model in Fig. 2. **b** Migration scheme for isolated nodes (no migration), star networks (all side nodes send a fraction $m$ of migrants towards the central node, which in turn sends a fraction $m/9$ towards each side node), and fully connected networks (every node sends a fraction $m/9$ of migrants towards each other node). **c** Experimental results showing the fraction of producers (computed as total number of producers over total number of cells for the whole network) as a function of time for the three network topologies, starting from three initial conditions. Each line corresponds to one 10-node network (or 10 isolated nodes)

does not take place inside the cell, but in the periplasmic space between the cell membrane and the cell wall, so most of the products (~99%) diffuse away, acting as a public good that can be used by any nearby cell. Cells that do not produce the enzyme can benefit from the public good without paying the cost of producing it (Fig. 1a)[48,49]. This system therefore contains three important interactions: cooperation among individuals of the producer strain, facilitation between the producer strain and the non-producer one, and competition for resources. Also, the

production of a public good gives rise to an Allee effect (i.e. the growth rate of the population is higher at intermediate densities than at low densities, due to the accumulation of public good). Because of this Allee effect, the population size does not diminish smoothly when conditions deteriorate, but undergoes a catastrophic collapse when the density of producers falls below a critical threshold[39,40]. This sudden collapse makes our experimental system ideal to study the stability of the metapopulation.

Additionally, this system presents both frequency- and density-dependent selection. When the population contains enough producers to make the public good plentiful, non-producers have the advantage of not paying the production cost, so they will increase in frequency. However, if too few producers are present the public good will be so scarce that the small amount imported by producers before it diffuses away will allow them to grow faster than the non-producers. These two effects create negative frequency-dependent selection, in which each strain is at a disadvantage when too frequent, and the population tends toward an intermediate fraction of both strains (Fig. 1a, left inset)[48,50]. This equilibrium fraction of producers in turn depends on the overall density of the population, giving rise to density-dependent selection: the same fraction of producers in a denser population entails more producer cells, hence more public good and greater advantage for the non-producers. Therefore, the equilibrium fraction of producers is lower in denser populations (Fig. 1a, right inset)[40].

This density-dependent selection may have important consequences in metapopulations, because asymmetric migration may lead to unequal densities in different nodes, and hence unequal fractions of producers. To study this effect, we compared isolated well-mixed populations to metapopulations with star topology, in which a central node is connected to 9 side nodes (star network in Fig. 1b). As a control, we also tested metapopulations with homogeneous topology (all nodes have the same number of neighbors) and symmetric migration (fully connected networks in Fig. 1b). In every time step, a fraction $m$ of the cells in each node migrate to neighboring nodes, distributing evenly among them (Fig. 1b, see Supplementary Methods for a step-by-step description of the protocol)[51]. This fraction $m$ of migrants is independent of the number of neighboring nodes, as is for example the case for organisms with a specialized dispersal stage. This migration scheme leads to asymmetric migration in heterogeneous networks, because the proportion of migrants traversing a link in each direction depends on the degree of the two connected nodes, with net migration flowing from the less connected to the most connected one (Fig. 1b, center). In the star network, net migration flows from the side nodes towards the center, which should lead to lower density on the sides and higher in the center, and therefore an increased fraction of producers in the sides and decreased in the center.

**Star networks have higher producer fraction in all nodes**. To determine how star networks affect the frequency of producers in the metapopulation, we performed experiments comparing isolated nodes, 10-node star networks, and 10-node fully connected networks. All populations underwent a daily dilution-migration-growth procedure: At the beginning of each day, all cultures were diluted in fresh medium by a factor 650. A fraction $m$ of the remaining cells in each node then migrated to neighboring nodes, distributing uniformly among them (Fig. 1b). The cells then grew for 23 h, until the next dilution-migration step. We chose a migration rate $m = 0.6$, which corresponds to around 6% per generation (cells undergo around 10 generations in every growth cycle).

We found that star networks had a higher overall fraction of producers. Regardless of the initial fraction, isolated populations and fully connected networks converge to having around 8% producers, while star networks show a two-fold increase over this value (Fig. 1c). A star network metapopulation structure therefore favors the public-goods producers in this system.

Furthermore, producer fraction increased in all nodes of the star networks, including the central one (Fig. 2a). As predicted, migration in the star network resulted in lower density for the side nodes and higher density for the central node, as compared to isolated populations (Fig. 2b). Yet both side and central nodes showed an increased fraction of producers (Fig. 2a). This increase in producer fraction is expected for the side nodes, as they experience a higher effective dilution rate due to asymmetric migration, thus leading to a decrease in cell density that favors producers due to the density-dependent selection (Fig. 1a). In contrast, the increased density in the central node should produce a decrease in the fraction of producers, as is indeed the case at the beginning of the experiment (Fig. 2a, days 1–5). However, the central node receives a large number of migrants from the side nodes. Therefore, once the fraction of producers in the side nodes is high enough, immigration into the central node increases its fraction of producers in spite of its high cell density. The heterogeneous network structure in our star network therefore increases the producer fraction in all nodes throughout the network.

To further understand these effects, we built a simple phenomenological model that incorporates negative frequency-dependent selection and density dependent selection. In this model, both strains grow logistically up to a common carrying capacity $K$. Their growth rates increase with the amount of available public good, which we assume to be proportional to the density of producers ($N_p$). We assumed Michaelis-Menten dynamics for this increase, with $k_M$ being the density of producers needed to produce enough public good to bring growth rate to half its maximum value. Because of the small fraction of sugars imported directly by producers, they benefit from an extra quantity $\varepsilon$ of public good[48]. Finally, producers pay a small cost $c$ for producing the public good (Fig. 2c). We used this model to simulate daily growth followed by 650× dilution and migration. This simple model successfully reproduces the increase in producer fraction that we observed experimentally (Fig. 2d, e). It also shows that the fraction of cooperators increases with the number of nodes of star networks and migration rate (Supplementary Figure 1).

**Star networks are less stable than isolated populations**. This phenomenological model also predicts the impact of heterogeneous structure on the metapopulation's stability in the face of deteriorating environments. In isolated populations, increasing the daily dilution factor eventually leads to a catastrophic collapse of the population[39,40]. Our model predicts that migration in star networks will facilitate this collapse, which will occur with milder dilution rates when migration rate is higher (Fig. 3a; in these figures fully connected networks give identical results to isolated populations, because symmetric migration has no effect when all nodes start from the same initial condition). This anticipated collapse in star networks happens because the lower density of the side nodes makes them incapable of sustaining the combined burden of dilution and net outward migration. Once the side nodes have collapsed, the central node receives no inward flux yet still has an outward flux of migrants, thus causing the central node to go extinct soon thereafter. Our model therefore predicts that star networks will go extinct in milder environmental

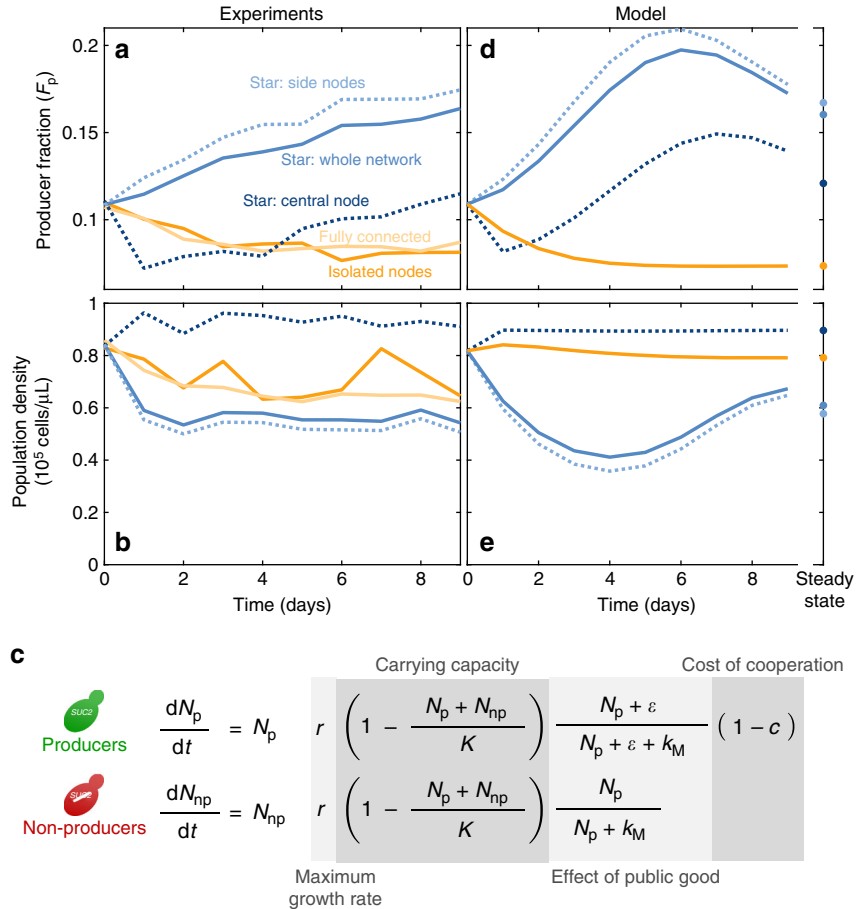

**Fig. 2** The increase in producer fraction in all nodes of a star network is captured by a simple phenomenological model. **a** Experimental results showing the time evolution of the fraction of producers in one star network on average (solid blue line), each type of node within the star network (dashed dark blue line for central nodes and dashed light blue line for side nodes), an isolated population (orange), and the fully connected network on average (pale yellow). **b** Same as **a**, but for the total density of cells. **c** Equations the phenomenological model. $N_p$, $N_{np}$ are the densities of producers and non-producers, respectively (in cells/μL); $r$ is the maximum growth rate; $K$ is the carrying capacity (common for both strains); $\varepsilon$ corresponds to the amount of enzyme imported by producers before the rest diffuses away (expressed in units of equivalent producer cells/μL, see Methods); $k_M$ is the amount of producers needed to produce enough enzyme to bring the growth of non-producers to half its maximum value; $c$ is the cost of producing the public good. Model parameters: $r = 0.5$ h$^{-1}$, $K = 90{,}000$ cells/μL, $c = 0.07$, $\varepsilon = 14$ cells/μL, $k_M = 26$ cells/μL, growth cycle 22 h, dilution factor 650 and migration rate $m = 0.6$ per cycle (~0.06 per generation). **d** Same as **a**, but as predicted by the model (points show the steady state after 100 cycles). Results for isolated nodes and fully connected networks overlap exactly. **e** Same as **b**, but as predicted by the model. Results for isolated nodes and fully connected networks overlap exactly

conditions than isolated populations, despite the higher producer fraction present in the star network.

To experimentally test this prediction of premature collapse, we compared the survival ability of isolated populations with that of the 10-node star networks over daily dilution factors from 400 to 2000. As predicted by the model, star networks collapse at lower dilution rates (Fig. 3b). For example, at a dilution rate of 1300 all four isolated populations survived, whereas none of the three star networks survived (Fig. 3b, inset). Therefore, isolated populations are better able to survive challenging environments than populations connected in a star network.

**Star networks are more resilient than isolated populations**. Previous reports have shown that a system close to a catastrophic collapse is less capable of recovering from harmful shocks[39,40,42,50]. In line with this, one would expect star networks to be less likely to recover from perturbations than isolated populations, since for a given dilution rate the heterogeneous networks are closer to the tipping point. We investigated this

prediction with the model, finding that it is only fulfilled in the immediate vicinity of the star network's tipping point. For most conditions, asymmetric migration increases the resilience of the metapopulation to a transient shock (in particular a transient decrease in population density) (Fig. 4a). To test whether this result depends on the nature of the perturbation, we investigated the metapopulation's resilience to both dilution shocks and growth rate shocks (i.e. decreasing the growth rate during one cycle). In both cases, we find higher resilience for star networks than for isolated nodes (Fig. 4b).

We tested experimentally this counterintuitive prediction or higher resilience in the metapopulation, by subjecting yeast populations to a growth-inhibiting high-salt environment (32 g/L) during one day. As predicted by the model, all three star networks recovered after the shock, while four of the five isolated populations went extinct (Fig. 4c). We therefore find that, despite being less able to survive sustained exposure to challenging environments, star networks are more resilient to transient environmental perturbations.

The surprising resilience of our star network is due to the increase in both density and producer fraction at the central node, which combine to increase the total number of producers present in the population. The salt shock leads to a smaller population in every node, which could take the density of producers below the threshold required for population survival. The increased number of producers allows the central node to survive perturbations that would drive isolated populations extinct. The side nodes of the network are not so resilient (because of their lower density), but they can be reseeded from the central node once the shock is over (Fig. 4d; note that the density of the central node still decreases during the first cycle after the shock, as a consequence of the outbound migration which is reseeding the 7side nodes, which is not yet compensated by any significant influx from them).

The effects we find on the stability and resilience of heterogeneous metapopulations do not require the mixture of producers and non-producers. We tested, both with the model and with experiments, that pure cultures of producers show the same trends: Star networks are less capable of surviving in challenging environments (Supplementary Figure 4), yet more resilient to transient perturbations (Supplementary Figure 5).

## Discussion

Our results highlight that stability and resilience may be determined by different factors in a complex system. In our heterogeneous metapopulation, stability depends on the weakest elements of the system—the side nodes—while resilience depends on the strongest one.

Our migration scheme links asymmetric migration with heterogeneous topology, because the proportion $m$ of outgoing migrants does not depend on the number of neighbors of their home node. This would be true for example for organisms with a specialized dispersal stage. However, migration patterns may differ across species. An opposite assumption would be that flux between two nodes is symmetric regardless of the topology. Many natural systems will probably be between these two extremes, and as long as heterogeneous connectivity produces some degree of asymmetric migration our qualitative results will be relevant. Future work that investigates alternative migration schemes will help disentangle the relative contributions of asymmetric migration and heterogeneous topology.

So far, we have used star networks as an example of a heterogeneous network. We have also investigated the generality of the results using our model for the complex heterogeneous topologies present within Barabási-Albert scale-free[52] networks. We find that the overall fraction of producers increases substantially (Supplementary Figure 6), similar to what we found experimentally in the star network. Also, the scale-free network shows decreased stability, although this effect is observed only at higher migration rates—for low migration rates, the stability of the network may actually increase (Supplementary Figure 7). Finally, we find that scale-free networks are more resilient than isolated populations (Supplementary Figures 8, 9). We also explored Watts-Strogatz small-world networks[53], where we find similar yet much weaker effects (Supplementary Figures 6–9). This weakness of the effects in small-world networks is consistent with our interpretation from our star network that it was the heterogeneity of the network that led to the observed effects (small-world networks are only slightly heterogeneous, with a variance in node degree of 0.84 in our simulations, compared with 8.9 for our scale-free network). In future work it would be interesting to explore a wider range of networks or even temporal networks[54]. Taken together, our results indicate that while our

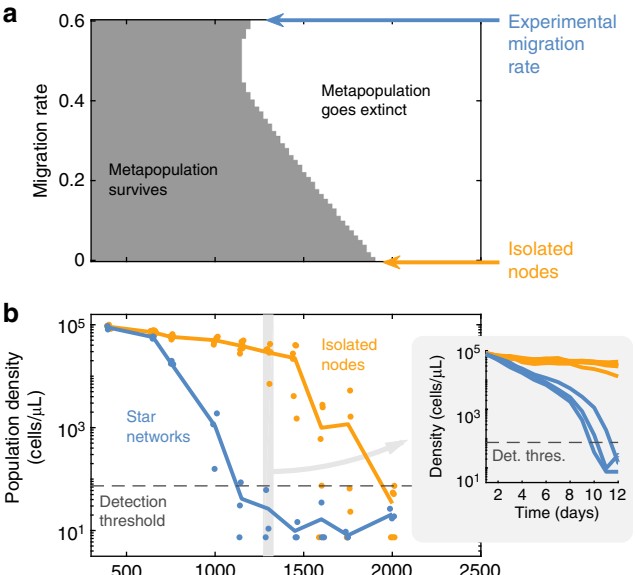

Fig. 3 Heterogeneous metapopulations are less able to survive challenging environments than isolated populations. **a** The gray area indicates the conditions (dilution factor and migration rate) in which the model predicts that a 10-node star network will survive after 1000 growth/dilution/migration cycles (note that $m = 0$ is also equivalent to isolated nodes). The rest of model parameters are the same as in Fig. 2. **b** Experimental equilibrium population densities for 10-nodes star networks (blue) and isolated nodes (yellow), as a function of dilution factor. Each dot is either one network or the average of a set of 10 isolated nodes; lines are the average. Inset: Experimental time series of population densities for dilution factor 1300, for star networks (blue) and isolated nodes (yellow). Each line is either one network or the average of a set of 10 isolated nodes. See Supplementary Figure 2 for the complete time series

conclusions from experiments with star networks can be extended to more general heterogeneous metapopulations, the strength of effects will depend on the network topology and its degree of heterogeneity.

The seemingly paradoxical result of lower distance to the tipping point but higher resilience to transient perturbations may be a common feature of asymmetry in networked systems: collapse in steady state is dictated by the weaker elements, while resilience to transient perturbations is dictated by the stronger ones. This is especially important given the pervasiveness of heterogeneous networks (such as scale-free networks) in nature, and may have parallels in other complex systems such as power grids or human populations[52].

## Methods

**Strains**. Both strain derive from haploid cells BY4741 (mating type a, EURO-SCARF). The producer strain, JG300B[48], has a wild-type SUC2 gene, and can therefore produce invertase for the breakdown of sucrose. It has a mutated HIS3 gene ($his3\Delta1$), therefore being a histidine auxotroph. It also expresses YFP constitutively. The non-producer strain, JG210C[48], has a deletion of the SUC2 gene, so it does not produce invertase. It has a wild-type HIS3 gene and expresses dTomato constitutively.

**Culture conditions**. Before every experiment, we picked a single colony of each cell type from a YPD agar plate (Teknova, Hollister, CA, USA; cat. no. Y1030), and cultured it in 5 mL of YNB + Nitrogen (Sunrise Science, CA, USA; cat no. 1501-250) + CSM-his (Sunrise Science, CA, USA; cat. no. 1001-100) supplemented with 2% glucose (Sigma-Aldrich/Millipore Sigma, St Louis, MO, USA; cat. no. G8270-1KG) and 8 µg/mL histidine (Sigma-Aldrich/Millipore Sigma, St Louis, MO, USA;

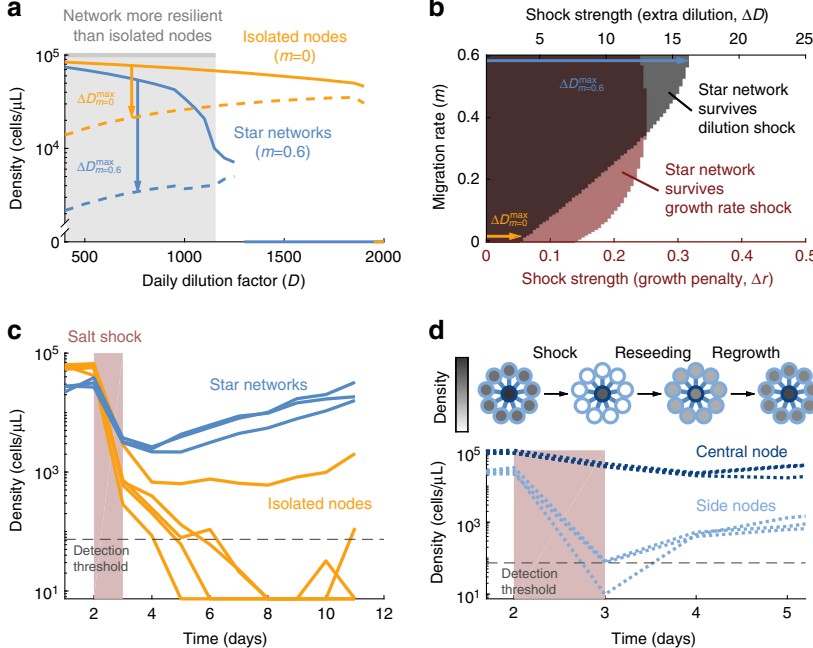

**Fig. 4** Star networks are more resilient to perturbations than isolated populations. **a** Bifurcation diagram for the metapopulation. Solid lines show the average population density in equilibrium as a function of dilution rate (yellow: isolated nodes; blue: 10-node star network). Dashed lines show the minimum density from which the metapopulation can recover. Arrows show the maximum dilution shock that each metapopulation can withstand ($\Delta D^{max}$), when daily dilution rate is $D = 750$. Gray area marks the region where the star network can withstand a greater shock than the isolated nodes. Model parameters are as in Fig. 2. See Supplementary Figure 3 for a more detailed version of this figure. **b** Model prediction for survival of a 10-node star network with daily dilution factor $D = 750$ after a perturbation, as a function of perturbation strength and migration rate. Rest of parameters are as in Fig. 2. Black: Shock corresponds to one cycle with increased dilution factor, $D' = 750^*\Delta D$. Red: Shock corresponds to one cycle with reduced growth $r' = 0.5 - \Delta r$. **c** Experimental results showing the time evolution of the average density in star networks (blue) and isolated populations (yellow). Each line corresponds to one 10-node network or to 10 isolated nodes. All populations are subject to a 750 dilution factor and are initially in equilibrium. They are then perturbed by a salt shock (32 g/L concentration of NaCl, shaded in red). **d** Top: Cartoon of star network recovering from a perturbation. Grayscale indicates population density in every node. Bottom: Same as **b**, but only for the star network and separating the center and side nodes

cat. no. 53319-25 G) in a 50-mL Falcon tube at 30 °C and 50% humidity with shaking at 250 r.p.m. for 24 h. We then mixed the two strains at different fractions and diluted them x100 in YNB + Nitrogen + CSM-his supplemented with 2% sucrose (Macron Fine Chemicals; cat. no. 8360-06), 0.001% glucose and 8 µg/mL histidine. We then incubated them for 24 h at 30 °C and 50% humidity with shaking at 250 r.p.m. (5 mL of culture in 50 mL Falcon tubes). The first day of the experiment we determined the fraction of each strain in each co-culture with flow cytometry, and mixed different co-cultures in order to achieve the desired starting fraction of producers for the experiment. This procedure ensured that the cells started in a physiological state characteristic of the co-culture of the two strains.

Experiments were performed in flat-bottom 96-well plates, with 200 µL of medium per well (YNB + CSM-his + 2% sucrose + 0.001% glucose + 8 µg/mL histidine). Plates were covered with Parafilm (Bemis Flexible Packaging, Neenah, WI, USA) to limit evaporation, and incubated for 23 h at 30 °C and 50% humidity with 800 r.p.m. shaking. After every incubation period, cells were diluted in fresh medium by the corresponding dilution factor, and migration was performed following the scheme described in the main text. Dilution and migration were performed at room temperature (~23 °C) and using two intermediate plates, to prevent pipeting of small volumes and ensure accurate dilutions. See Supplementary Information for a step-by-step description of the dilution-migration protocol.

**Measurements**. Total density of every subpopulation was measured at the end of each growth cycle by measuring the optical density at 600 nm in a plate reader (Varioskan Flash, Thermo Fisher Scientific). Fraction of producers was also determined at the end of each growth cycle by flow cytometry (Macs Quant VYB, Miltenyi Biotec, Bergisch Gladbach, Germany). Experimental results reported in Figs. 1–4 were replicated in two separate experiments. Experiments reported in Supplementary Figures 4, 5 were performed once.

**Model**. We simulated the system using a model that reproduces the discrete cycles of the experiment: The populations grow during each cycle governed by the equations shown in Fig. 2c (we used Matlab's function ode45 to solve the differential equations numerically). Then, all populations are divided by the dilution

rate, and migration is performed as detailed in Fig. 1b. Then the next growth cycle is simulated.

To determine the model's parameters, we experimentally determined the parameters $r = 0.5 \, h^{-1}$ and $K = 90,000$ cells/µL. From previous works we know that $c < 0.1$, $\varepsilon \approx k_M$ and that the lag phase of yeast should be between 1 and 4 h[48]. Within these constraints, we manually fitted the exact values of $c$, $\varepsilon$ and the growth cycle duration to reproduce the trends shown in Fig. 2, as well as the collapse dilution rates shown in Fig. 3. We found a good agreement for $c = 0.07$, $\varepsilon = 14$ cells/µL, $k_M = 26$ cells/µL and a growth cycle of 22 h (i.e. a lag phase of 1 h). A more detailed fit was unnecessary, given that this simple phenomenological model does not capture the quantitative details of the system.

Note that $\varepsilon$ is in units of the density of producer cells that should exist in the culture to bring the concentration of public good to a level that matches the amount that each producer cell keeps for itself.

To perform the resilience tests, we first let the metapopulation reach a stable state. If this stable state presented oscillations (see Supplementary Figure 3), we chose a cycle in which population density was minimum (given that we expect the metapopulation to be least resilient at this point). Then we perturbed the metapopulation during a single cycle, either by imposing an additional dilution factor $\Delta D$ (so for one cycle the dilution factor was $D' = D^*\Delta D$), or by reducing the growth rate by $\Delta r$ (so for one cycle the growth rate was $r' = r - \Delta r$). After the perturbation cycle all parameters went back to normal, and the simulation continued until the metapopulation either recovered or went extinct.

**Code availability**. All computer programs are available from the corresponding author upon request.

**Data availability**. All the data are available from the corresponding author upon request.

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

## Acknowledgements

We are grateful for useful discussions with Kirill S. Korolev, the members of the Gore lab and the members of the Barabási Lab. This work was supported by the DARPA BRICS program (Program Code: TRS-01 under Contract No. HR0011-15-C-0091), NIH New Innovator Award (DP2 AG044279), NSF CAREER Award (PHY-1055154), a Sloan Research Fellowship (BR2011-066), the Pew Scholars Program (2010-000224-007), the Paul Allen Family Foundation, an EMBO Postdoctoral Fellowship (Grant ALTF 818-2014), a Human Frontier Science Foundation Postdoctoral Fellowship (Grant LT000537/2015), NSFC (Grants No. 61375120 and No. 61533001), China Scholarship Council (No. 201406010195).

## Author contributions

Am.L. and J.G. conceived the project; An.L. and A.P-E. performed the experiments and the analyzed experimental data; An.L., A.P-E. and Am.L. performed modeling on star

and fully connected networks; Am.L. performed modeling on complex networks; all authors wrote, reviewed and commented the paper.

## Additional information

**Competing interests:** The authors declare no competing interests.

