## [Peer Review File · Nature Communications]

Reviewers' comments:

Reviewer #1 (Remarks to the Author):

This is an excellent paper combining an elegant series of experiments and theory to show how the pattern (symmetry) of migration in population networks mediates the likelihood of collapse under density independent mortality (dilution), and resilience to perturbations. The results are nontrivial—they run counter to often made assumptions about migration and rescue in metapopulations—because of the interplay between density and frequency dependence in the dynamics of this yeast public good 'game'. The counter-intuitive results are clearly explained, thanks in part to the mathematical model of the system. The conclusion that network structure may benefit some measures of persistence (recovery from perturbations) and not others (e.g. nonlinear collapse) is important.

General comment:

I would like to see the model applied to different network structures. Only star networks are examined here with the experiments, but one is left wondering how general the findings are, or rather, how specific they are (qualitatively and quantitatively) to this 9-node star network structure. The model could be used to look at a range of star networks sizes and configurations, but it can also be used to look at other network structures; the star network is a rather specific and unnatural structure. For example, the paper starts with reference to dendritic networks, which are a common ecological example of asymmetric flows. The impact of the paper would be much greater if some additional results could be provided.

Specific Comments:

The paper is framed as a metacommunity study but it is actually a metapopulation system. A metacommunity is defined "as a set of local communities that are linked by dispersal of multiple potentially interacting species" (Leibold et al. 2004). These experiments are focused on two strains of the same yeast species, *S. cerevisiae*. I think it is important not to relax the metacommunity definition so that it merges with that of a metapopulation. I would urge the authors to correct this through-out.

Line 50: Papers cited at this point don't reflect recent progress with designed metapopulation/metacommunity experiments. A few from my own lab:

Gilarranz, L. J., Rayfield B., Liñán-Cembrano, G., Bascompte, J., Gonzalez, A. (2017) Effects of network modularity on the spread of perturbation impact in experimental metapopulations. *Science* 357: 199-201.

You might cite Bell and Gonzalez (2011) as an example of a metapopulation study focused on evolution and survival of *S. cerevisiae* in stressful environments.

Bell, G. and Gonzalez, A. (2011) Adaptation and Evolutionary Rescue in a metapopulation during environmental deterioration. *Science* 332: 1327-1330.

Low-Decarie et al. (2015) is also an example of experiment that studies evolution and persistence in spatially explicit multispecies metacommunities exposed to environmental degradation.

Low-Decarie, E, Kolber, M, Homme, P., Lofano, A, Gonzalez, A, Bell, G. (2015) Community rescue in metacommunities adapting to environmental stress. *PNAS* 112: 14307-14312.

Tekwa et al. (2017) also study cooperation and metapopulation persistence in networked habitats. Tekwa EW, Nguyen D, Loreau M, Gonzalez A. 2017 Defector clustering is linked to cooperation in a

pathogenic bacterium. Proc. R. Soc. B 284: 20172001.

Signed,
Andy Gonzalez

Reviewer #2 (Remarks to the Author):

This study examines the impact of meta-population structure on the ability of a two-strain microbial community to withstand different kinds of disturbance regime. The authors use a toy model and the well-established experimental system of invertase producing and non-producing yeast cells, set up as isolated populations or 10-population star networks. At low population density within a population, the system will crash because too few producers are present to supply sufficient sucrose digestion. The core results of the paper comment on the resilience of isolated versus meta-populations to sustained shocks (stronger dilution factor), versus single shocks (transient stronger dilution factor, or salt shock).

The model and experiments indicate that sustained shocks to the population size (stronger dilution factor) can more easily cause a metapopulation to crash than they can cause an isolated population to crash. Specifically, the threshold dilution factor at which the metapopulations crash out is lower than the threshold dilution factor at which isolated populations crash out. The authors interpret this effect to be due to the fact that isolated populations have a steady state density (and therefore critical sustainable dilution factor) that is intermediate between that of the peripheral sub-groups (lower equil. density) and central sub-groups (higher equil. density) of the metapopulations. Peripheral sub-groups of the metapopulations are more easily wiped out by dilution events than isolated populations, and the central, higher-density sub-groups of the metapopulations gradually bleed out as they re-supply the peripheral sub-groups with new members each growth cycle.

For single shocks to population size, contingent on the background dilution factor, the metapopulations are more resilient than isolated sub-populations. This is again because of the difference in equilibrium population density between the highly-connected and weakly-connected nodes of the metapopulation. When the weakly connected peripheral nodes collapse, they can be re-seeded by the highly-connected, higher population density node and eventually recover; when isolated populations collapse, they stay collapsed. This holds so long as the shock event is not big enough to eliminate the core highly-connected node of the metapopulation, which has a higher equilibrium density and therefore higher tolerance to single shocks than an isolated population.

This is generally an elegant study, and I think it will be well received as a commentary on the importance of metapopulation structure and migration on population stability against perturbation. It is appreciated that the strong dependence of the results on the Allee effect of the system was clearly emphasized in the discussion. I did not see problems with the methodology but did find a few issue that warrants attention in a revision. There is a cooperation problem at play in the model and experimental populations, i.e. competition between invertase producers and non-producers. However, in Figures 3 and 4 and the discussion that goes with them (the core novel results), these two strains are no longer analyzed as distinct entities, and it is not clear if it is important that both of them are present for the main findings of the paper re: resilience and disturbance regime. Intuitively it seems like all of the core results on metapopulation versus isolated population stability should hold for a system where only the producer is present, so long as the Allee effect is in place. Does the Allee effect here depend on the presence of the non-producer? If so this was not clearly established in the introduction and early figures. Detailed elaboration to this effect would be very helpful.

Reviewer #3 (Remarks to the Author):

In their article "Asymmetric migration decreases stability..." the authors use a combination of experimental work and mathematical modelling to study the effect of asymmetric migration on metacommunity composition and resilience. I like the combination of empirical and mathematical work, and their conclusion that asymmetric migration reduces tolerance to certain perturbations while increasing tolerance to others is an interesting one. However, I think the manuscript requires further work before it is of the standard required by Nature Communications.

The first part, investigating the role of community structure on the success of a producer strain is well posed, if not very surprising. The authors impose a particularly extreme migration scheme that leads to a large number of low density metacommunities that favour producers. But how dependent is this phenomenon on the strict migration scheme imposed? Having constructed a mathematical model that well captures their experimental scheme I recommend the authors make more use of it – what happens if you have variability in migration rate, or better still, a slightly more variable asymmetric network structure (e.g. small world)? I think these are relatively simple but important extensions if the authors want to make sweeping statements about the role of asymmetric migration.

The second part - suggesting somewhat of a trade-off between tolerance to challenging environments vs transient perturbations – is more interesting. However, while I don't necessarily think they are wrong, this section needs further work before it can be considered convincing. The authors have dropped their comparison of star vs fully connected networks - as such I don't feel they can make statements about the role of asymmetric migration in particular. Is it asymmetric migration specifically that drives this increased / decreased tolerance, or just migration in general? On similar lines, the authors have two processes at work here – asymmetric migration and public good dynamics, both of which will be contributing to community stability. Does asymmetric migration only affect community stability when these specific public goods dynamics are also at play, or is this a broader phenomenon? The authors state that they would not observe the same behaviour without the density dependent selection, however they have not shown this.

We thank the Editor and referees for their careful reading of the manuscript. Below (in blue) we describe how we have addressed the questions (in black). We have included in our submission a version of the manuscript with "track changes" activated, to aid in the reviewing process.

Reviewer #1 (Remarks to the Author):

This is an excellent paper combining an elegant series of experiments and theory to show how the pattern (symmetry) of migration in population networks mediates the likelihood of collapse under density independent mortality (dilution), and resilience to perturbations. The results are nontrivial—they run counter to often made assumptions about migration and rescue in metapopulations—because of the interplay between density and frequency dependence in the dynamics of this yeast public good 'game'. The counter-intuitive results are clearly explained, thanks in part to the mathematical model of the system. The conclusion that network structure may benefit some measures of persistence (recovery from perturbations) and not others (e.g. nonlinear collapse) is important.

General comment:

I would like to see the model applied to different network structures. Only star networks are examined here with the experiments, but one is left wondering how general the findings are, or rather, how specific they are (qualitatively and quantitatively) to this 9-node star network structure. The model could be used to look at a range of star networks sizes and configurations, but it can also be used to look at other network structures; the star network is a rather specific and unnatural structure. For example, the paper starts with reference to dendritic networks, which are a common ecological example of asymmetric flows. The impact of the paper would be much greater if some additional results could be provided.

We have incorporated four new supplementary figures where we study the predictions of the model with different topologies, including different star networks, Watts-Strogatz small-world and Barabási-Albert scale-free networks. Our results indicate that the main conclusions of the paper are robust, holding in general for heterogeneous networks with asymmetric migrations. These additional results can be found in lines 169-172, 179, and 210 of the main text and supplementary figures S1, S2, S3 and S4.

Specific Comments:

The paper is framed as a metacommunity study but it is actually a metapopulation system. A metacommunity is defined "as a set of local communities that are linked by dispersal of multiple potentially interacting species" (Leibold et al. 2004). These experiments are focused on two strains of the same yeast species, *S. cerevisiae*. I think it is important not to relax the metacommunity definition so that it merges with that of a metapopulation. I would urge the authors to correct this through-out.

We feel that the effects found in our synthetic system may be relevant both in the context of metapopulations and metacommunities, but we agree that perhaps the term metapopulation is more natural. As requested, we have changed this throughout the text (including the title of the paper).

Line 50: Papers cited at this point don't reflect recent progress with designed metapopulation/metacommunity experiments. A few from my own lab:

Gilarranz, L. J., Rayfield B., Liñán-Cembrano, G., Bascompte, J., Gonzalez, A. (2017) Effects of network modularity on the spread of perturbation impact in experimental metapopulations. *Science* 357: 199-201.

You might cite Bell and Gonzalez (2011) as an example of a metapopulation study focused on evolution and survival of *S. cerevisiae* in stressful environments.

Bell, G. and Gonzalez, A. (2011) Adaptation and Evolutionary Rescue in a metapopulation during environmental deterioration. *Science* 332: 1327-1330.

Low-Decarie et al. (2015) is also an example of experiment that studies evolution and persistence in spatially explicit multispecies metacommunities exposed to environmental degradation.

Low-Decarie, E, Kolber, M, Homme, P., Lofano, A, Gonzalez, A, Bell, G. (2015) Community rescue in metacommunities adapting to environmental stress. *PNAS* 112: 14307-14312.

Tekwa et al. (2017) also study cooperation and metapopulation persistence in networked habitats. Tekwa EW, Nguyen D, Loreau M, Gonzalez A. 2017 Defector clustering is linked to cooperation in a pathogenic bacterium. *Proc. R. Soc. B* 284: 20172001.

Thank you for pointing this out. We have added the suggested references (lines 58-60)

Signed,
Andy Gonzalez

Reviewer #2 (Remarks to the Author):

This study examines the impact of meta-population structure on the ability of a two-strain microbial community to withstand different kinds of disturbance regime. The authors use a toy model and the well-established experimental system of invertase producing and non-producing yeast cells, set up as isolated populations or 10-population star networks. At low population density within a population, the system will crash because too few producers are present to supply sufficient sucrose digestion. The core results of the paper comment on the resilience of isolated versus meta-populations to sustained shocks (stronger dilution factor), versus single shocks (transient stronger dilution factor, or salt shock).

The model and experiments indicate that sustained shocks to the population size (stronger dilution factor) can more easily cause a metapopulation to crash than they can cause an isolated population to crash. Specifically, the threshold dilution factor at which the metapopulations crash out is lower than the threshold dilution factor at which isolated populations crash out. The authors interpret this effect to be due to the fact that isolated populations have a steady state density (and therefore critical sustainable dilution factor) that is intermediate between that of the peripheral sub-groups (lower equil. density) and central sub-groups (higher equil. density) of the metapopulations. Peripheral sub-groups of the metapopulations are more easily wiped out by dilution events than isolated populations, and the central, higher-density sub-groups of the metapopulations gradually bleed out as they re-supply the peripheral sub-groups with new members each growth cycle.

For single shocks to population size, contingent on the background dilution factor, the metapopulations are more resilient than isolated sub-populations. This is again because of the difference in equilibrium population density between the highly-connected and weakly-connected nodes of the metapopulation. When the weakly connected peripheral nodes collapse, they can be re-seeded by the highly-connected,

higher population density node and eventually recover; when isolated populations collapse, they stay collapsed. This holds so long as the shock event is not big enough to eliminate the core highly-connected node of the metapopulation, which has a higher equilibrium density and therefore higher tolerance to single shocks than an isolated population.

This is generally an elegant study, and I think it will be well received as a commentary on the importance of metapopulation structure and migration on population stability against perturbation. It is appreciated that the strong dependence of the results on the Allee effect of the system was clearly emphasized in the discussion. I did not see problems with the methodology but did find a few issues that warrant attention in a revision.

There is a cooperation problem at play in the model and experimental populations, i.e. competition between invertase producers and non-producers. However, in Figures 3 and 4 and the discussion that goes with them (the core novel results), these two strains are no longer analyzed as distinct entities, and it is not clear if it is important that both of them are present for the main findings of the paper re: resilience and disturbance regime. Intuitively it seems like all of the core results on metapopulation versus isolated population stability should hold for a system where only the producer is present, so long as the Allee effect is in place. Does the Allee effect here depend on the presence of the non-producer? If so this was not clearly established in the introduction and early figures. Detailed elaboration to this effect would be very helpful.

Indeed, the effect does not depend on the presence of the non-producer, and this result was not sufficiently explored in our previous version. We have performed new simulations and experiments with pure cooperator populations, and found that our results also hold in these pure cultures. We have incorporated these new results in lines 227-231 and supplementary figures S7 and S8.

Reviewer #3 (Remarks to the Author):

In their article "Asymmetric migration decreases stability..." the authors use a combination of experimental work and mathematical modelling to study the effect of asymmetric migration on metacommunity composition and resilience. I like the combination of empirical and mathematical work, and their conclusion that asymmetric migration reduces tolerance to certain perturbations while increasing tolerance to others is an interesting one. However, I think the manuscript requires further work before it is of the standard required by Nature Communications.

The first part, investigating the role of community structure on the success of a producer strain is well posed, if not very surprising. The authors impose a particularly extreme migration scheme that leads to a large number of low density metacommunities that favour producers. But how dependent is this phenomenon on the strict migration scheme imposed? Having constructed a mathematical model that well captures their experimental scheme I recommend the authors make more use of it – what happens if you have variability in migration rate, or better still, a slightly more variable asymmetric network structure (e.g. small world)? I think these are relatively simple but important extensions if the authors want to make sweeping statements about the role of asymmetric migration.

As suggested, we have used the model to explore different topologies where migration rate varies over asymmetric Watts-Strogatz small-world and Barabási-Albert scale-free networks. We find that our results are quite robust to the other topologies, holding in general for heterogeneous networks with various asymmetric migrations. These additional results can be found in lines 169-172, 179, and 210 of the main text and supplementary figures S1, S2, S3 and S4.

The second part - suggesting somewhat of a trade-off between tolerance to challenging environments vs transient perturbations – is more interesting. However, while I don't necessarily think they are wrong, this section needs further work before it can be considered convincing. The authors have dropped their comparison of star vs fully connected networks - as such I don't feel they can make statements about the role of asymmetric migration in particular. Is it asymmetric migration specifically that drives this increased / decreased tolerance, or just migration in general?

Thanks for pointing this out. We now clarify why we drop this comparison: For the model, these dynamics are indistinguishable from those of the isolated populations. This will always be the case in simulations with homogeneous networks (or with symmetric migration): The initial condition is the same for all nodes, so either homogeneous networks or symmetric migration will result on having the same number of individuals of each type going in and out of each node. Therefore, any theoretical result with symmetric migration or homogeneous connectivity will always be identical to the one for isolated populations. We now clarify this in the text (lines 179-181).

These theoretical results match with our experimental results in Figure 1, and we have now added the fully connected network in Figure 2. In the following experiments we dropped the control with fully connected networks due to space constraints in our experimental plates. As discussed above, we have also incorporated the new simulations in figures S1-S4, which further clarify the role of migration.

On similar lines, the authors have two processes at work here – asymmetric migration and public good dynamics, both of which will be contributing to community stability. Does asymmetric migration only affect community stability when these specific public goods dynamics are also at play, or is this a broader phenomenon? The authors state that they would not observe the same behaviour without the density dependent selection, however they have not shown this.

We agree that the exact role of public-good dynamics was insufficiently explored in the previous version. We have performed simulations to test the stability and resilience of a metapopulation with pure cooperator populations, finding the same qualitative results. We have also performed new experiments with pure cooperator cultures that confirm these results (i.e. heterogeneous metapopulations are less capable of surviving at high dilution rates, but more capable of withstanding a transient shock). These results are presented in lines 227-231, and in figures S7 and S8.

We also realize that our previous version might imply that all our results were due to density-dependent selection. We have deleted the misleading paragraph around line 232.

Reviewers' comments:

[Reviewer #1 confidentially expressed satisfaction with responses to their comments]

Reviewer #2 (Remarks to the Author):

The authors have done a good job in addressing the concerns of the referees.

Reviewer #3 (Remarks to the Author):

My key concerns during the initial review were

1. The authors made sweeping statements about the role of asymmetric migration while only investigating one very restrictive community structure, alongside a more general lack of clarity between the role of asymmetric migration vs migration in general.
2. It was unclear to what extent these results were dependent on the public goods dynamics as opposed on heterogeneous migration alone.

The authors have done a good job to address point 2, however point 1 needs further work.

The authors have now included simulations for small-world and scale-free networks (figs S2-4) and state in their main text that these support their results that asymmetric migration systems have higher fractions of cooperators, are less resilient to challenging environments, and are more resilient to transient shocks.

However, the figures themselves do not seem to match these statements. In particular in figure S3, as the authors state in the legend, the SW and SF networks each seem to fare **better** than the isolated nodes at intermediate levels of migration (in contrast to the star network). Meanwhile, the legend of fig S4 states there is **no difference** in ability to withstand growth penalties for SW networks compared with isolated nodes. The figure appears to show a small difference, but also seems to have sampled parameter space sparsely and unevenly so taken alongside the legend I'm not clear whether the authors consider this to be real. Altogether then I'm still not convinced the authors can fairly make the broad statement 'asymmetric migration decreases stability but increases resilience...'.

At the very least these points need further discussion in the text, and the text/title need to be amended to clarify these results hold specifically for star networks and cannot be assumed to be true for asymmetric migration in general. Even better though would be if the authors could better disentangle this link between level of migration asymmetry and stability/resilience. What is driving the initial increase in stability for the SW/SF networks for example?

Thank you for your consideration. We are glad to see that the reviewers find that most of their concerns are now addressed. Reviewer #3 still has one concern, and we have revised the text accordingly (see more detailed comments below).

Reviewers' comments:

[Reviewer #1 confidentially expressed satisfaction with responses to their comments]

Reviewer #2 (Remarks to the Author):

The authors have done a good job in addressing the concerns of the referees.

Reviewer #3 (Remarks to the Author):

My key concerns during the initial review were

1. The authors made sweeping statements about the role of asymmetric migration while only investigating one very restrictive community structure, alongside a more general lack of clarity between the role of asymmetric migration vs migration in general.
2. It was unclear to what extent these results were dependent on the public goods dynamics as opposed on heterogeneous migration alone.

The authors have done a good job to address point 2, however point 1 needs further work.

Thank you for your new revision. We agree that our text was somewhat inconsistent, and we have revised it (see below)

The authors have now included simulations for small-world and scale-free networks (figs S2-4) and state in their main text that these support their results that asymmetric migration systems have higher fractions of cooperators, are less resilient to challenging environments, and are more resilient to transient shocks.

However, the figures themselves do not seem to match these statements. In particular in figure S3, as the authors state in the legend, the SW and SF networks each seem to fare **better** than the isolated nodes at intermediate levels of migration (in contrast to the star network). Meanwhile, the legend of fig S4 states there is **no difference** in ability to withstand growth penalties for SW networks compared with isolated nodes. The figure appears to show a small difference, but also seems to have sampled parameter space sparsely and unevenly so taken alongside the legend I'm not clear whether the authors consider this to be real. Altogether then I'm still not convinced the authors can fairly make the broad statement 'asymmetric migration decreases stability but increases resilience...'.

At the very least these points need further discussion in the text, and the text/title need to be amended to clarify these results hold specifically for star networks and cannot be assumed to be true for asymmetric migration in general. Even better though would be if the authors could better disentangle this link between level of migration asymmetry and stability/resilience. What is driving the initial increase in stability for the SW/SF networks for example?

Thanks for pointing this out. We agree that our previous description of Figure S3 was a bit oversimplified. As you observed, for intermediate levels of migration, both SW and SF networks are slightly better able to survive challenging environments. Now we discuss this point in our revised version (lines 215-217 and figure caption of Figure S3).

For transient shocks shown in Fig. S4, we have improved our analysis in two ways: First, following your constructive comment, we have run more simulations to obtain a finer phase space. Second, we have studied more topologies, splitting our old Figure S4 in two figures (S4 and S5). We find that the increase in resilience is stronger in scale-free networks than in small-world ones. This difference may be due to scale-free networks being more heterogeneous than small-world ones (variance in node's degree is 8.9 and 0.84, respectively). Apart from our results for star network with high heterogeneity, we believe that the stability/resilience of networked populations deserves more systematic investigations both experimentally and analytically for general heterogeneous networks, which is beyond the main scope of the current manuscript, and this has also been pointed out in our new version.

Finally, we have reduced the generality of our claims, now stating that asymmetric migration decreases stability but increases resilience in the star networks we studied. We indicate that these effects can also be found in other heterogeneous networks like SF, but the SW and SF networks can also be more stable than the isolated one for challenging environments under intermediate level of migrations.

In summary, we have made the following changes:

- We have corrected the figure captions of Figs. S3 and S4.
- We have split the old Fig. S4 in two figures: new S4 and S5, which also have higher resolution.
- We have edited the text, describing the star topology in the abstract (lines 27-32), mentioning only star networks in the "Results" section, and moving our discussion about other topologies to the discussion (lines 211-227).

The reviewers did not raise any concerns in this round or revision. We would like to thank them for their work and their very useful comments in previous rounds.